# Tuning the Curing Efficiency of Conventional Accelerated Sulfur System for Tailoring the Properties of Natural Rubber/Bromobutyl Rubber Blends

**DOI:** 10.3390/ma15238466

**Published:** 2022-11-28

**Authors:** Marek Pöschl, Shibulal Gopi Sathi, Radek Stoček

**Affiliations:** Centre of Polymer Systems, Tomas Bata University in Zlín, Tr. TřidaTomášeBati 5678, 760 01 Zlín, Czech Republic

**Keywords:** rubber, curing, bismaleimide, tensile strength, compression set, Diels–Alder reaction

## Abstract

The state of cure and the vulcanizate properties of a conventional accelerated sulfur (CV) cured 50/50 blend of natural rubber (NR) and bromobutyl rubber (BIIR) were inferior. However, this blend exhibits a higher extent of cure with remarkable improvements in its mechanical properties, particularly the tensile strength, modulus and hardness after curing with a combination of accelerated sulfur and three parts per hundred rubber (phr) of a bismaleimide (MF_3_). Moreover, with the use of 0.25 phr of dicumyl peroxide (DCP) along with the CV/MF_3_ system, the compression set property of the CV-only cured blend could be reduced from 68% to 15%. The enhanced compatibility between NR and BIIR with the aid of bismaleimide via the Diels–Alder reaction was identified as the primary reason for the improved cure state and the mechanical properties. However, the incorporation of a certain amount of bismaleimide as a crosslink in the NR phase of the blend, via a radical initiated crosslinking process by the action of DCP, is responsible for the improved compression set properties

## 1. Introduction

Natural rubber is a non-polar, highly unsaturated elastomer and is chemically called cis-1, 4-poly (isoprene). Because of the high level of cis-content (99.9%) in the microstructure of NR, it forms crystallites upon stretching. The ability of NR to crystallize upon stretching gives additional strength to NR and is technically referred to as strain hardening or strain-induced crystallization. As a result, even the vulcanized gum NR can exhibit a tensile strength in the range of 20–25 MPa [1,2,3]. Moreover, NR has excellent green strength (uncured rubber strength) and tackiness which make NR stick to itself and also enhances its ability to bond with metals. Because of these qualities, the majority of NR is consumed in the tire sector of the rubber industry. However, the highly unsaturated backbone of NR weakens its resistance to atmospheric oxygen and ozone. As a result, it undergoes degradation if proper care has not been taken. To overcome these shortcomings of NR, it is frequently blended with other elastomers [4,5,6,7,8]. However, developing a proper curing system to make a compatible blend of NR with other elastomers like BIIR is still a challenge.

It is well-known that, being an unsaturated elastomer, NR is widely cured with sulfur and a suitable type of accelerator to speed up the curing process. Depending upon the accelerator to sulfur (A/S) ratio, three types of accelerated sulfur systems are commonly employed in the rubber industries to create a crosslink structure in rubber materials. They are the conventional vulcanization (CV) system, semi-efficient vulcanization (SEV) system and the efficient vulcanization (EV) system. In the conventional vulcanization system, the sulfur content is generally kept higher than the accelerator dose. In addition to the accelerated sulfur curing systems, the NR can also be cured using peroxides such as dicumyl peroxide (DCP) [9,10,11]. On the other hand, the brominated version of poly (isobutylene-co-isoprene) rubber (butyl rubber) is a random copolymer of isobutylene with 1–3 mol% of isoprene units. Here, the isoprene units which are randomly distributed in the BIIR can be employed for the curing reaction with the accelerated sulfur system. Moreover, BIIR can also be cured using ZnO and can produce thermally stable carbon-carbon crosslinks in the cured network [12,13]. Datta et al. studied the effect of a sulfur donor, maleimide and phenolic resin curing system on NR/BIIR and reported that co-vulcanization between NR and BIIR occurs via interfacial bond formation [14]. However, unlike NR, BIIR does not undergo crosslinking with peroxide; rather, it undergoes degradation at the curing temperature due to macro-radical fragmentation [15,16].

It is well-known that the compression set (CS) is an important benchmark for developing elastomeric gaskets and seals. Rubber compounds with a lower percentage of the CS can perform better as sealing materials. One of the reasons for the failure of the gasket is its inability to snap back to its original thickness after prolonged compressive stress at the given service temperature due to the loss of its elasticity or memory. If the network created during the curing process of a rubber compound is strong enough and can maintain a good elasticity even after compressive stress in a hot atmosphere, its CS value will be naturally low. In the present article, an attempt has been made to create carbon-carbon crosslinks in the NR phase of its 50/50 blend with BIIR to improve the cure compatibility and compression set by using a combination of bismaleimide and DCP, which is the novelty of this work.

## 2. Materials

Natural rubber (SVR CV60, Mooney viscosity ML (1 + 4) at 100 °C: 60 ± 5) and Bromobutyl rubber (Exxon bromobutyl 2224, Mooney viscosity ML (1 + 8, 125 °C: 46 ± 5) were used as the base elastomers. A combination of 75% N, N’-meta phenylene dimaleimide and a 25% blending agent (Maleide F) from Krata Pigment, Tambov, Mentazhnikov, Russia was used as the co-curing agent. The chemical structure of Maleide F is represented in Figure 1. Other curing agents such as sulfur, n-cyclohexyl-2-benzothiazole sulfenamide (CBS), stearic acid, zinc oxide and dicumyl peroxide (DCP) were purchased from Sigma-Aldrich, Czech Republic.

### 2.1. Compounding

Table 1 indicates the formulations of the compounds with designations. All the compounds were prepared using an internal mixer (Brabender Plastograph, GmbH & Co. KG, Duisburg, Germany) with a chamber volume of 50 cm^3^. The component rubbers were masticated separately for 2 min at 50 °C under 50 rpm and then mixed together for a further minute. To this, the ZnO, stearic acid and MF were added, and the mixing was continued for another 2 min with 25 phr of carbon black (N330). Finally, the sulfur, CBS and DCP were added and mixed for an additional minute. The rubber compound was discharged and homogenized using a two-roll mill. It was then molded into sheets with a thickness of 2 mm by applying a constant force of 200 N using a compression molding heat press (LaBEcon 300, Fontijne Presses, Delft, The Netherlands) based on the Rheometer cure data at 170 °C.

### 2.2. Characterization

#### 2.2.1. Cure Properties

A moving die rheometer (MDR-3000, Mon Tech, Buchen, Germany) was used to measure the cure properties such as the maximum torque: M_H_, minimum torque: M_L_ and the difference, M_H_-M_L_: ΔM, the scorch time: TS_2_, and the optimum cure time: T_90_, of the rubber compounds as per ASTM D 5289 at 170 °C for 1 h. The cure rate index (CRI), a measure of the rate of curing, was calculated using Equation (1).
CRI = 100/(T_90_ − TS_2_)(1)

#### 2.2.2. Tensile Properties

The tensile properties of the vulcanizates (S2 type specimen with 2 mm thickness) were measured using a universal testing machine (Testometric M350, Testometric Company, Ltd., Rochdale, UK) at a crosshead speed of 500 mm/min as per ISO 37. The results were reported at an average of six tested specimens.

#### 2.2.3. Hardness Testing

A Shore-A hardness tester (Bareiss Durometer, Oberdischingen, Germany), as per ASTM D 2240, was used to measure the hardness of the cured samples. Six readings were taken from different areas of the sample by applying a constant indentation pressure for 3 s and reporting the average value.

#### 2.2.4. Compression Set

The compression set (CS) was measured as per DIN ISO 815. Cylindrical samples with 6 mm thickness and 13 mm diameter were subjected to a 25% compression. The compressed samples were then kept in an air-oven at 100 °C for 22 h. After the specified testing time, the samples were taken out and allowed to cool for 30 min at room temperature and the thickness was measured. The CS was calculated using Equation (2).
(2)$$CS(\%)=\frac{t_{0}-t_{f}}{t_{0}-t_{s}}\times 100$$
where *t*_0_ and *t_f_* are the original and final thickness of the specimen and *t_s_* is the thickness of the spacer used.

## 3. Results and Discussion

### 3.1. Curing Behavior and the Mechanical Properties of the NR, BIIR and NR/BIIR Blend with a CV System

Depicted in Figure 2 is the curing curve of the NR/BIIR blend with a CV system and the cure characteristics are displayed in Table 2. The extent of curing in terms of delta torque (ΔM) was only 3.64 dNm. Moreover, the blend shows a sharp declination in the rheometer torque after 14 min of curing due to reversion. This low extent of curing indicates that CV alone is not an effective curing system to co-cure NR and BIIR, which may be due to the cure-rate incompatibility of CV between the two elastomers. The reversion behavior that is observed in the curing curve of the blend indicates that substantial polysulfidic crosslinks have been created in the NR phase. To substantiate this assumption, the curing of NR and BIIR were separately studied using the CV system by maintaining the same acceleration to sulfur ratio. Their curing curves and the cure characteristics obtained at 170 °C for 1 h are also depicted in Figure 2 and Table 2, respectively.

The extent of curing in NR with CV was higher than in NR/BIIR with CV. However, the onset of reversion starts just after 5 min of curing due to the breakage of the polysulfidic crosslinks developed in the cured network [17,18]. On the other hand, the extent of curing was the lowest in the BIIR-CV system. This is probably due to the low amount of the cure site (isoprene) units in BIIR to generate more sulfur-based crosslinks in its network. It is interesting to note that no reversion was observed in BIIR-CV until the end of the given curing time. One of the reasons for this might be the generation of some carbon-carbon crosslinks in its cured network by the action of ZnO present in the CV system.

The tensile properties such as tensile strength (TS), elongation at break (EB), modulus and hardness of NR-CV, BIIR-CV and NR/BIIR-CV were evaluated after curing at 170 °C up to their respective T_90_, and the results are reported in Table 3. NR-CV exhibited the highest TS, maybe because of the inherent strain-induced crystallization behavior of NR, and also due to the flexible nature of the polysulfidic crosslinks in its vulcanized network. The TS of BIIR-CV was considerably lower compared to NR-CV. This is probably due to a small number of sulfidic crosslinks in the cured network of BIIR-CV. However, the blend of NR/BIIR-CV exhibited the least TS and modulus. This may be due to the low extent of curing owing to the cure rate incompatibility of CV with NR and BIIR.

In addition to the tensile properties, the compression sets of these compounds were also measured and the results are reported in Table 4. It is a well-known fact that the compression set of a rubber compound gives an idea of the strength of a cured network because this property is measured under a hot atmosphere with a static load for a specified time [19]. As expected, the compression set of NR-CV was around 61.6% compared to BIIR-CV which was about 23.5%. The breaking and subsequent rearrangements of the polysulfidic crosslinks into di-and mono sulfidic within the network of NR-CV by the application of heat and pressure can be considered as one of the reasons for its high compression set value. On the other hand, the formation of thermally stable C-C crosslinks in the cured network of BIIR-CV by the action of ZnO present in the CV system might be the reason for its low compression set value [20]. However, it was expected that the CS of NR/BIIR-CV would come somewhere between the CS of NR-CV and BIIR-CV because of the possibilities of creating both the sulfidic as well as the C-C crosslinks during the curing of NR/BIIR with CV. Surprisingly, the CS of NR/BIIR-CV was the highest (64.5%) compared to NR-CV and BIIR-CV. It has been observed that the curing of NR-CV is relatively fast (high CRI) compared to BIIR-CV. Therefore, in NR/BIIR-CV, the NR phase of the blend is crosslinked a little faster and produces sulfidic crosslinks in the network. This may quickly enhance the overall viscosity of the system and also leads to a phase separation between NR and BIIR. As a result, the ZnO in the CV system may be incapable of producing sufficient carbon-carbon crosslinks in the BIIR phase of the blend. This may reduce the overall network strength and elasticity of NR/BIIR-CV and it therefore exhibited a high compression set value. However, to check whether any carbon-carbon crosslinks have been created in the BIIR phase of NR/BIIR-CV in the later stage of curing, its CS was also measured from the samples prepared after 45 min of press curing, and the result is also reported in Table 4.

For comparative purposes, the CS value of 45 min cured NR-CV and BIIR-CV were also reported. The CS value of the 45-min cured NR-CV was 21% decreased compared to the T_90_ + 5 min cured one, even if its curing torque at T_90_ + 5 min was much higher than the torque value generated at 45 min of curing. Similarly, the CS of 45-min cured NR/BIIR-CV was around 25% lower than the CS of its T_90_ + 5 min cured counterpart. An around 18% reduction in the CS value was also noticed in the 45-min cured samples of BIIR-CV. The longer curing time (45 min) applied to all of these compounds may result in the conversion of the majority of the initially generated polysulfidic crosslinks into di- and mono-sulfidic crosslinks. That is, the network of the 45-min cured samples contains more di- and mono-sulfidic crosslinks than the network of the T90 + 5 min cured samples. Since the di-and mono-sulfidic crosslinks are thermally more stable than the polysulfidic crosslinks, the CS measured from the 45-min cured samples is naturally low. It is interesting to note that the CS of NR/BIIR-CV was still significantly higher than BIIR-CV even if both of them exhibit the same curing torque beyond 45 min of curing. This confirms that no additional C-C crosslinks have been generated in NR/BIIR-CV by extending the curing time beyond its T_90_.

### 3.2. Curing Behavior and the Mechanical Properties of NR/BIIR-CV with Bismaleimide and Dicumyl Peroxide

It has been reported that the cure state of NR with the CV system can be improved with the use of a small amount of bismaleimide via the Diels–Alder (DA) reaction [18]. It has also been reported that the curing efficiency of ZnO in BIIR can be enhanced with the use of a small amount of bismaleimide along with the ZnO [20]. In light of this information, 3 phr of bismaleimide (MF_3_) was used in combination with the CV system to co-cure NR and BIIR. The curing curves of NR/BIIR with CV and with CV/MF_3_ at 170 °C are depicted in Figure 3. 

Surprisingly, the state of cure in terms of Δtorque in NR/BIIR-CV was improved by 165% in the presence of MF. Moreover, the reversion that was observed in NR/BIIR-CV after 14 min of curing could be eliminated when the same mixture was cured along with 3 phr of MF. This curing behavior raised three theoretical possibilities to justify the enhanced state of cure. One of the possibilities might be the DA reaction between the MF and the diene formed in-situ from the NR segments by the action of the CV system [18]. The DA reaction between the diene released from BIIR by the action of ZnO and the maleimide moieties of MF might be the second possibility [20]. However, the DA reaction between the diene formed in-situ from the NR phase with one end of the maleimide moieties of MF and the diene formed in-situ from BIIR with another end of the MF as depicted in stage I of Figure 4 is considered as the third possibility [8,21]. If the first two possibilities are dominant, they may boost the incompatibility due to phase separation between NR and BIIR because of the selective crosslinking of one phase of the blend. On the other hand, if the DA reaction mentioned as a third possibility is dominant, it may enhance the compatibility between NR and BIIR. To check whether the compatibility is improved or not, the tensile properties of NR/BIIR-CV were measured after curing with 3 phr of MF and the results are also displayed in Table 3. The TS of NR/BIIR-CV was improved by 60% after the incorporation of 3 phr of MF. In addition to this, approximately a 97% improvement in the modulus at 100% elongation and a 47% improvement in the shore-A hardness were also observed.

The improved tensile properties confirm the fact that the compatibility between NR and BIIR has been enhanced after curing the blend with a combination of CV and bismaleimide. To further check the strength of the cured network that has been created in NR/BIIR via the above-mentioned compatibilization reaction with the help of CV/MF_3_, the CS property of NR/BIIR-CVMF_3_ was measured and reported in Table 4. For the purposes of comparison, the CS properties of NR-CVMF_3_ and BIIR-CVMF_3_ were also measured and depicted in the same Table. The CS value that has been observed in NR/BIIR-CV could be reduced from 64.5% to 29.4% after curing with 3 phr of MF. One of the reasons for this could be the formation of thermally stable bismaleimide-based adduct-type crosslinks in its vulcanized network. The TS and CS values of NR/BIIR-CVMF_3_ reveal the fact that the NR phase has substantial sulfidic crosslinks even after curing with MF. This might be one of the reasons for its enhanced TS. On the other hand, these sulfidic crosslinks are responsible for the relatively high CS value in NR/BIIR-CVMF_3_ compared to BIIR-CVMF_3_.

To further improve the network strength in NR/BIIR-CVMF_3_, an attempt has been made to introduce some carbon-carbon crosslinks in the NR phase via MF with the use of dicumyl peroxide (DCP). Represented in Figure 5 are the curing curves of NR/BIIR-CVMF_3_ with different contents of DCP and their cure characteristics are also reported in Table 2. 

The Δ torque of NR/BIIR-CVMF_3_ gradually increased as the content of DCP increased. This indicates that the overall network strength of the blend was enhanced. Moreover, the speed of the curing reaction increased as the content of DCP increased. As a result, the T_90_ of NR/BIIR-CVMF_3_ could be reduced from 15.7 min to just 7.15 min when cured with 1 phr of DCP. It is interesting to note that NR/BIIR-CVMF_3_ and NR/BIIR-CVMF_3_D_0.25_ exhibited a plateau-type curing behavior without any reversion until the end of the given curing time. However, as the content of DCP increased to 0.5 phr, a declination in the cure curve was observed after 19 min of curing with a reversion of around 3.6% at the end of the given curing time. As the content of DCP rose to 1 phr, the point of reversion started a little earlier (at 15 min) and the blend exhibited about 4.7% reversion at the end of curing. It is a well-known fact that BIIR undergoes degradation when it is trying to crosslink with peroxide [16]. Therefore, the observed reversion in the curing curves of NR/BIIR-CVMF_3_ beyond 0.25 phr of DCP may be due to the degradation of the BIIR segments in the blend by the action of DCP.

To understand how the changes that have been developed in the network by the action of DCP would affect the vulcanizate properties of the blend, the mechanical properties of NR/BIIR-CVMF_3_ were evaluated as a function of the DCP contents, and the results are represented in Figure 6a–e. The TS was reduced to 33% and the EB was reduced to 38% when NR/BIIR-CVMF_3_ was cured with 0.25 phr of DCP. However, approximately a 71% improvement in the modulus was noticed after curing the same mixture with 0.25 phr of DCP. These tensile properties indirectly indicate that some additional crosslinks have been formed in NR/BIIR-CVMF_3_ by the action of DCP. As a result, the overall stiffness in the cured network increases which would naturally enhance the modulus but reduce the elongation and hence the TS. It is interesting to note that the CS of NR/BIIR-CVMF_3_ was reduced to around 50% when it was cured with 0.25 phr of DCP. The CS value of NR/BIIR-CVMF_3_ was further reduced to 59% as the content of DCP rose to 1 phr. This implies that DCP creates a thermally stable network composed of radically initiated bismaleimide-based crosslinks in the NR phase in addition to the co-crosslinked bismaleimide bridges between NR and BIIR as depicted in stage II of Figure 4. Here, the bismaleimide bridges formed via the DA reaction enhance the compatibility between NR and BIIR, and the bismaleimide connected between the chains of NR with the help of DCP impart additional network strength in the system. As a result, the DCP cured NR/BIIR-CVMF_3_ exhibits a low CS value at elevated temperatures.

## 4. Conclusions

The rheometer curing analysis undoubtedly revealed that the efficiency of accelerated sulfur is boosted and thereby enhances the state of cure in the NR/BIIR blend in the presence of 3 phr bismaleimide. The huge improvement in tensile strength after curing the blend with the accelerated sulfur/bismaleimide combination supported the fact that the compatibility between NR and BIIR was enhanced by the action of bismaleimide. The Diels–Alder reaction between the in-situ formed dienes from NR and BIIR at the time of curing with either end of the maleimide moieties of the bismaleimide was identified as one of the probable reasons for the enhanced compatibility. The significantly low compression set (27%) of NR/BIIR-CVMF_3_ compared to the compression set (64.5%) of NR/BIIR-CV proves that the bismaleimide crosslinks formed in the network are thermally stable. The radically connected bismaleimide crosslinks in the NR phase of the blend (NR/BIIR-CVMF_3_) by the action of DCP further improve its network strength. As a result, the compression set of NR/BIIR-CV could be reduced from 64.5% to just 14.9% after curing with a combination of 3 phr bismaleimide and 0.25 phr DCP.

## Figures and Tables

**Figure 1 materials-15-08466-f001:**
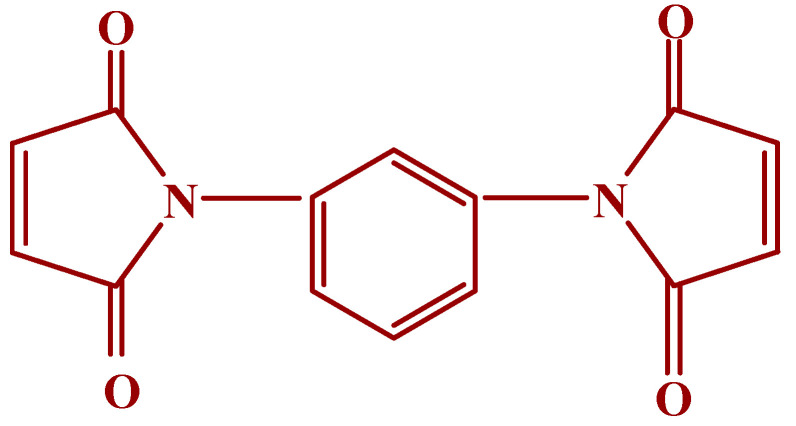
N, N’-meta phenylene dimaleimide.

**Figure 2 materials-15-08466-f002:**
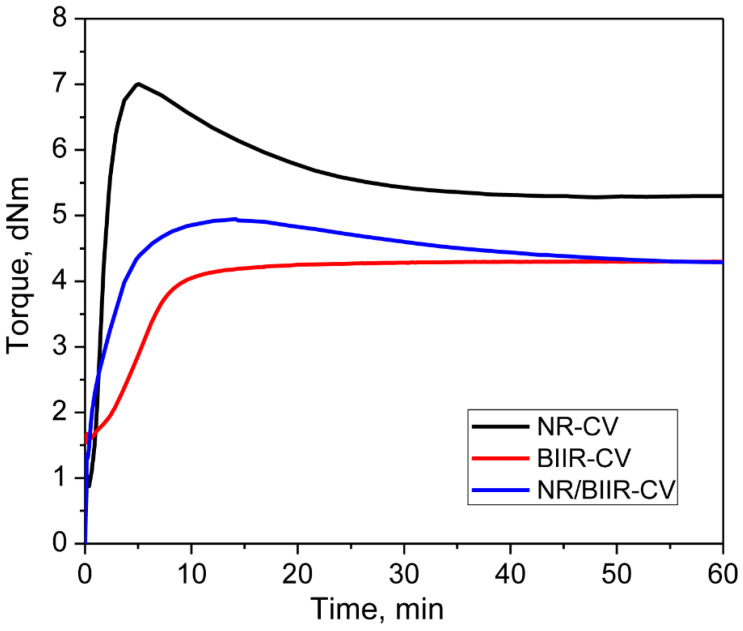
Curing curves of NR-CV, BIIR-CV and NR/BIIR-CV.

**Figure 3 materials-15-08466-f003:**
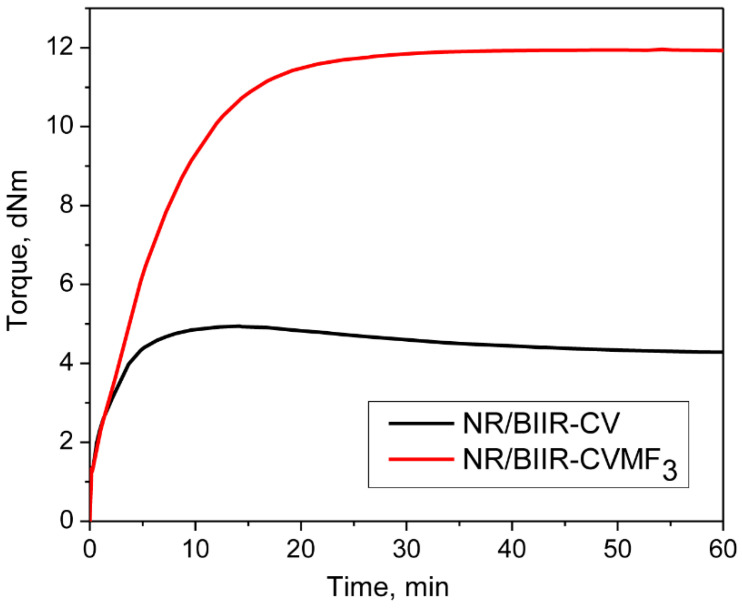
Curing curves of NR/BIIR-CV and NR/BIIR-CVM3 at 170 °C for 1 h.

**Figure 4 materials-15-08466-f004:**
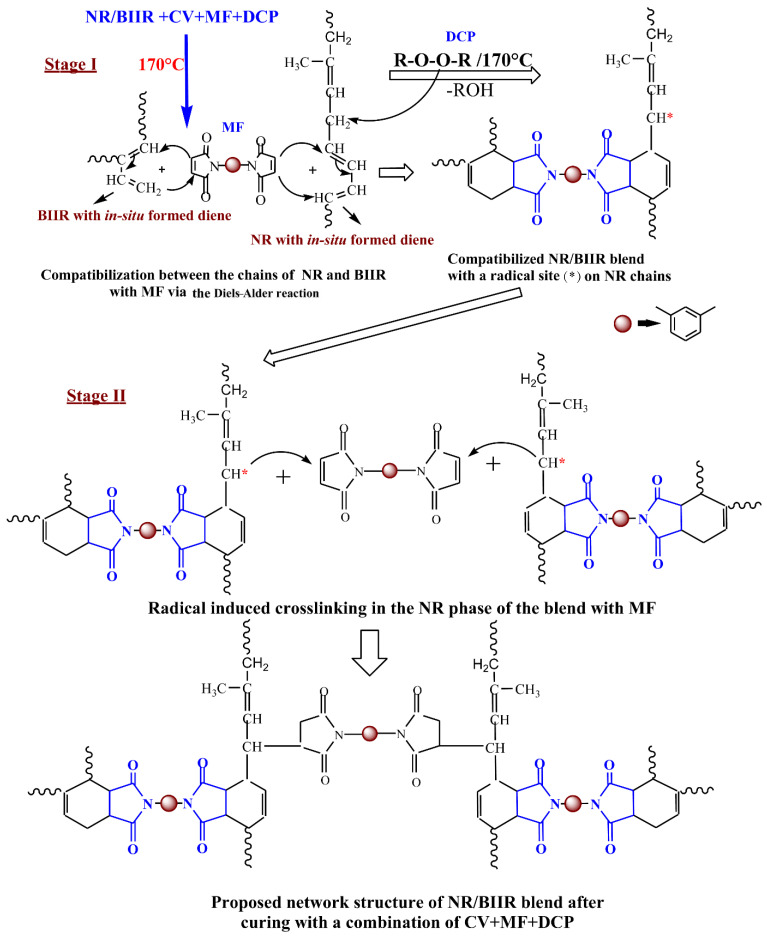
Plausible crosslinking mechanism responsible for the enhanced compatibility and low compression set properties of the NR/BIIR blend with CV/MF/DCP.

**Figure 5 materials-15-08466-f005:**
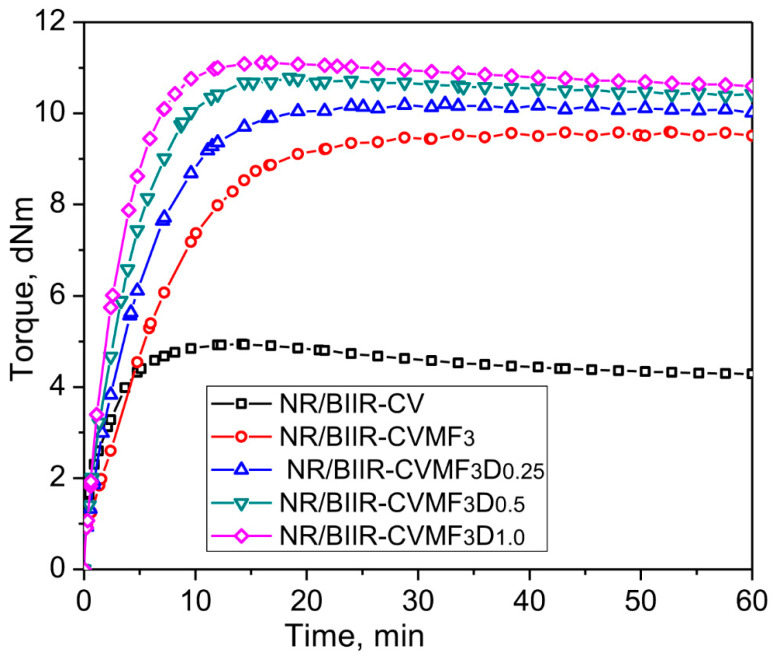
Curing curves of NR/BIIR-CVMF3 with different contents of DCP at 170 °C for 1 h.

**Figure 6 materials-15-08466-f006:**
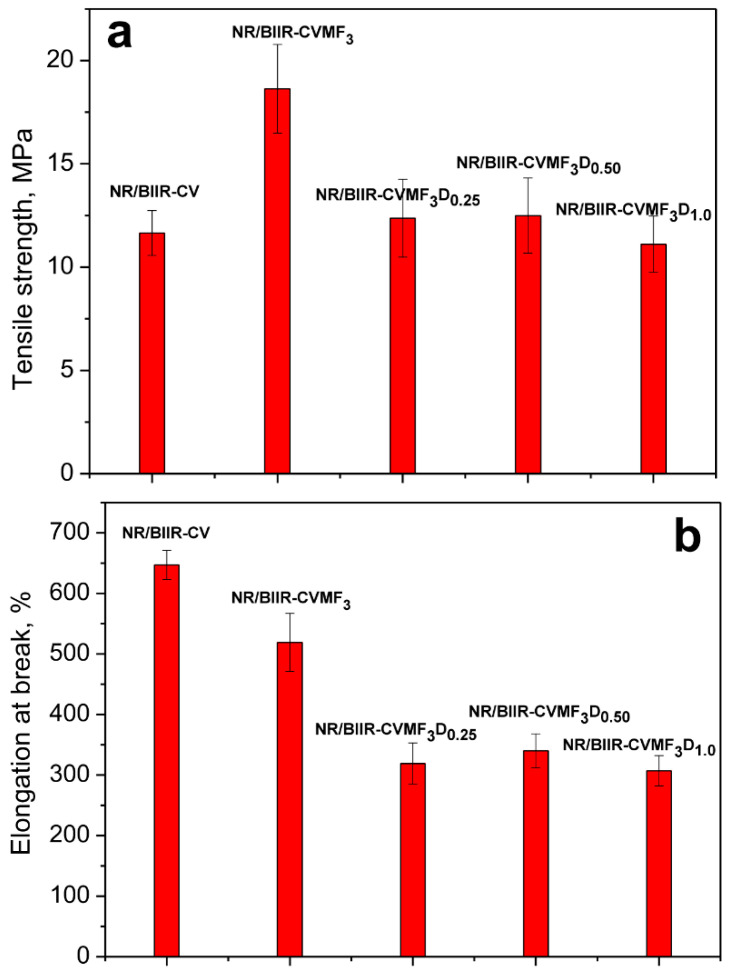
Mechanical properties of NR/BIIR-CVMF3 with different contents of DCP (**a**) Tensile strength; (**b**) Elongation at break; (**c**) Modulus at 100% elongation; (**d**) Shore-A Hardness; (**e**) Compression set.

**Table 1 materials-15-08466-t001:** Mix compositions.

Mix ID *	NR	BIIR	ZnO	St.Acid	Sulfur	CBS	MF	DCP
NR/BIIR-CV	50	50	5	1	1.25	0.25	-	-
NR-CV	100	-	5	2	2.5	0.5	-	-
NR-CVMF_3_	100	-	5	2	2.5	0.5	3	-
BIIR-CV	-	100	5	-	0.5	0.1	-	-
BIIR-CVMF_3_	-	100	5	-	0.5	0.1	3	-
NR/BIIR-CVMF_3_	50	50	5	1	1.25	0.25	3	-
NR/BIIR-CVMF_3_-D_0.25_	50	50	5	1	1.25	0.25	3	0.25
NR/BIIR-CVMF_3_-D_0.5_	50	50	5	1	1.25	0.25	3	0.5
NR/BIIR-CVMF_3_-D_1.0_	50	50	5	1	1.25	0.25	3	1

* 25 phr of N330 CB was added to all the mixes.

**Table 2 materials-15-08466-t002:** Cure characteristics of the mixes at various temperatures.

Mix ID	ML (dNm)	MH (dNm)	ΔM (dNm)	TS2 (min)	T90 (min)	CRI (min^−1^)
NR/BIIR-CV	1.30	4.94	3.64	0.93	6.33	18.51
NR-CV	0.88	7.00	6.12	1.07	3.08	49.75
NR-CVMF_3_	0.75	11.33	10.58	1.29	6.44	19.41
BIIR-CV	1.59	4.30	2.71	4.28	9.72	18.38
BIIR-CVMF_3_	1.69	8.21	5.76	2.28	8.30	16.61
NR/BIIR-CVMF_3_	0.98	9.33	8.35	1.54	15.70	7.06
NR/BIIR-CVMF_3_-D_0.25_	0.93	10.22	9.29	0.90	11.05	9.85
NR/BIIR-CVMF_3_-D_0.5_	1.00	10.78	9.78	0.72	8.88	12.25
NR/BIIR-CVMF_3_-D_1.0_	0.91	11.11	10.20	0.52	7.15	15.08

**Table 3 materials-15-08466-t003:** Tensile properties and hardness.

Sample ID	Tensile Strength (MPa)	Elongation at Break (MPa)	Modulus at 100% (MPa)	Hardness (Shore A)
NR/BIIR-CV	11.65 ± 1.09	647 ± 24	0.76 ± 0.05	36 ± 1
NR-CV	17.81 ± 1.71	653 ± 61	0.84 ± 0.08	38 ± 1
BIIR-CV	14.62 ± 1.26	834 ± 33	0.86 ± 0.39	35 ± 1
NR/BIIR-CVMF_3_	18.63 ± 2.14	519 ± 48	1.50 ± 0.09	53 ± 1

**Table 4 materials-15-08466-t004:** Compression set properties.

Sample ID	Curing Time (min)	Compression Set (%)
NR/BIIR-CV	T_90_ + 5	64.5 ± 2.05
	45	47.9 ± 3.16
NR-CV	T_90_ + 5	61.6 ± 0.25
	45	48.4 ± 1.42
NR-CVMF_3_	T_90_ + 5	37.7 ± 0.75
BIIR-CV	T_90_ + 5	23.5 ± 0.78
	45	18.8 ± 0.70
BIIR-CVMF_3_	T_90_ + 5	11.7 ± 0.70
NR/BIIR-CVMF_3_	T_90_ + 5	29.4 ± 1.20

## Data Availability

The data presented in this study are available on request from the corresponding author.

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
