# Peer review of "Tuning the Curing Efficiency of Conventional Accelerated Sulfur System for Tailoring the Properties of Natural Rubber/Bromobutyl Rubber Blends"

_materials, 2022, doi:10.3390/ma15238466_

Round 1

Reviewer 1 Report

The.manuscript.is.well.written.and.study.is.done.systematically.

1.Few.highlight.the.novelty.of.the.present.work.as.it.issimilar.to.theprevious.works.done.

2.The.authors.may.show the mechanism of the curing in the presence of DCP and absence of DCP separately.

3.It.is.known.that the presence of DCP.can degrade.BIIR. Please.explain why did.you.use DCP in the curing system?

4.Also,the,authors.have.to.address.the.plagiarism.in.themanuscript.

Author Response

The.manuscript.is.well.written.and.study.is.done.systematically.

Thanks for the encouraging words about our work.

Comment1.Few.highlight.the.novelty.of.the.present.work.as.it.issimilar.to.theprevious.works.done.

Response: This work is not similar to our previous works rather it is a continuation of our previous works. From the previous studies, we understood that a combination of accelerated sulphur and bismaleimide can co-crosslink NR and BIIR (CIIR) via the Diels-Alder reaction. However, in this work, we introduced a combination of bismaleimide and DCP to create some additional carbon-carbon crosslinks via bismaleimide in the NR segments of the already co-crosslinked NR/BIIR blends by the action of accelerated-sulphur and bismaleimide with the aid of some amount of DCP which is the novelty of this work.

Comment 2.The.authors.may.show the mechanism of the curing in the presence of DCP and the absence of DCP separately.

Response:  Yes, You can see the curing mechanism without DCP on the left-hand side of the reactions depicted in Stage I of Figure 4. And, the right-hand side of stage I and stage II constitute the curing mechanisms in the presence of DCP.

Comment 3. It.is.known.that the presence of DCP.can degrade.BIIR. Please.explain why did.you.use DCP in the curing system?

Response: Yes, we do agree with the reviewer's statement, of course, DCP can degrade BIIR. However, in this study, we deliberately used DCP as a radical initiator particularly to connect the bismaleimide between the NR segments (some parts of which is already been in a co-cross-linked state with BIIR) to reduce the overall compression set of the blend. To avoid degradation of the BIIR segments, we limit the content of DCP to a minimum concentration of 0.25phr. Our result was positive and we could significantly reduce the compression set of the blend with the use of DCP.

Comment 4. Also, the authors.have.to.address.the.plagiarism.in.themanuscript.

Response: Yes, you may feel some similarity in connection with our previous articles because it is the continuation of our previous studies. However, all the data in this article are original and highly repeatable. As suggested by the reviewer, the content of repetition has been taken care in the revised script.

Reviewer 2 Report

The authors reported an effective vulcanization system (accelerated bismaleimide-sulfur-DCP system) for curing natural rubber/bromobutyl rubber (NR/BIIR) blends. The vulcanization behavior and mechanical properties of NR/BIIR blends under accelerated bismaleimide-sulfur-DCP system and conventional vulcanization (CV) system were compared and investigated in detail. In particular, the curing mechanism of accelerated bismaleimide-sulfur-DCP system for NR/BIIR blends was well elucidated. However, there are some minor errors in the manuscript which are needed to be addressed before publication.

1. In Abstract, CV stands for conventional vulcanization, which should be kept consistent with the term in the Introduction section. 

phr should be given definition when it appears at the first time. For convenience, the usage amount of vulcanization agents can be ignored, as it is not necessary for understanding.

2. In Introduction, the literatures about NR/BIIR system are insufficient, especially, the vulcanization system for NR/BIIR system.

3. In Results and Discussions, 

“The reversion behaviour that is observed in the curing curve of the blend indicates that whatever the network formed after the curing is mainly of sulfidic crosslinks (mono-, di-and poly-) which may be primarily located in the NR phase.” Should be revised.

…… the accelerated to sulfur ratio as 0.2 as per the formulations described in the Table 1” should be revised.

“Set value” should  be “compression set value”.

“Thermal stability” is a special term that refers to the thermal degradation properties of polymer chains. Therefore, in Line 175, “This may subside the overall thermal stability of NR/BIIR-CV and therefore exhibited a high set value” should be revised.

The X axis of Figure 6a is hidden. and the unit of the Modulus at 100% elongation is missing in Figure 6c.

Author Response

The authors reported an effective vulcanization system (accelerated bismaleimide-sulfur-DCP system) for curing natural rubber/bromobutyl rubber (NR/BIIR) blends. The vulcanization behavior and mechanical properties of NR/BIIR blends under an accelerated bismaleimide-sulfur-DCP system and conventional vulcanization (CV) system were compared and investigated in detail. In particular, the curing mechanism of accelerated bismaleimide-sulfur-DCP system for NR/BIIR blends was well elucidated. However, there are some minor errors in the manuscript which are needed to be addressed before publication.

Response: Thank you for understanding our work and giving suggestions for the improvements

Comment 1. In the Abstract, CV stands for conventional vulcanization, which should be kept consistent with the term in the Introduction section.

phr should be given a definition when it appears for the first time. For convenience, the usage amount of vulcanization agents can be ignored, as it is not necessary for understanding.

Response:  It has been modified as per the suggestion.

Comment 2. In the Introduction, the literatures about NR/BIIR system are insufficient, especially, the vulcanization system for NR/BIIR system.

Response: Literature is scanty concerning the vulcanization studies of NR/BIIR blends. One article reverent to this work has been added now as reference No 14

Comment 3. In Results and Discussions, 

“The reversion behaviour that is observed in the curing curve of the blend indicates that whatever the network formed after the curing is mainly of sulfidic crosslinks (mono-, di-and poly-) which may be primarily located in the NR phase.” Should be revised.

“ …… the accelerated to sulfur ratio as 0.2 as per the formulations described in the Table 1” should be revised.

“Set value” should  be “compression set value”.

“Thermal stability” is a special term that refers to the thermal degradation properties of polymer chains. Therefore, in Line 175, “This may subside the overall thermal stability of NR/BIIR-CV and therefore exhibited a high set value” should be revised.

The X axis of Figure 6a is hidden. and the unit of the Modulus at 100% elongation is missing in Figure 6c.

Response: Thank you very much for indicating the mistakes. All the corrections have been addressed and highlighted in the revised script as suggested.

Reviewer 3 Report

1. The title is very long and tedious, so it should be revised.

2. The introduction should be improved because it can not reflect the recent progress in the field. The novelty can be emphasised on.

3. In Fig.6, the unit name was missing.

4. In 2.1, 2.2 , the expression of unit should be improved. For example, "50cc",  "2mm", "22hrs",  etc.

Author Response

Comment 1. The title is very long and tedious, so it should be revised.

Response: Slightly modified the title.

Comment 2. The introduction should be improved because it can not reflect the recent progress in the field. The novelty can be emphasised on.

Response: Modified as suggested and added the novelty of this work.

Comment 3. In Fig.6, the unit name was missing.

Response: Done

Comment 4. In 2.1, 2.2 , the expression of unit should be improved. For example, "50cc",  "2mm", "22hrs",  etc.

Response: It has  been modified now